# The Galactic Interstellar Medium Has a Preferred Handedness of Magnetic Misalignment

**Zhiqi Huang** [1,2]

1   School of Physics and Astronomy, Sun Yat-sen University, 2 Daxue Road, Tangjia, Zhuhai 519082, China; huangzhq25@mail.sysu.edu.cn
2   CSST Science Center for the Guangdong-Hongkong-Macau Greater Bay Area, Sun Yat-sen University, Zhuhai 519082, China

**Abstract:** The Planck mission detected a positive correlation between the intensity ($T$) and $B$-mode polarization of the Galactic thermal dust emission. The $TB$ correlation is a parity-odd signal, whose statistical mean vanishes in models with mirror symmetry. Recent work has shown, with strong evidence, that local handedness of the misalignment between the dust filaments and the sky-projected magnetic field produces $TB$ signals. However, it remains unclear whether the observed global $TB$ signal is caused by statistical fluctuations of magnetic misalignment angles or whether some parity-violating physics in the interstellar medium sets a preferred misalignment handedness. The present work aims to make a quantitative statement about how confidently the statistical fluctuation interpretation is ruled out by filament-based simulations of polarized dust emission. We use the publicly available DUSTFILAMENTS code to simulate the dust emission from filaments whose magnetic misalignment angles are symmetrically randomized and construct the probability density function of $\xi_p$, a weighted sum of the $TB$ power spectrum. We find that the Planck data have a $\gtrsim 10\sigma$ tension with the simulated $\xi_p$ distribution. Our results strongly support the idea that the Galactic filament misalignment has a preferred handedness, whose physical origin is yet to be identified.

**Keywords:** dust; ISM: magnetic fields; ISM: structure—cosmic background radiation; polarization; submillimeter: diffuse background

## 1. Introduction

The cosmic microwave background (CMB) is the most powerful cosmological probe known to date. Observations of the CMB temperature and polarization anisotropies by the Planck satellite and many other experiments are in good agreement with the standard picture of a Lambda cold dark matter (ΛCDM) universe that begins with an inflationary epoch [1–5]. Yet, the $B$-mode polarization in the CMB, which is regarded as the smoking gun of early-universe inflation, has not been observed [6–8]. The Galactic foreground and, in particular, the thermal emission from interstellar dust grains has become one of the major obstacles to achieving higher-precision measurements of the $B$-mode polarization of the CMB [9–11]. Thus, future CMB science crucially depends on the ability to understand the physics and the statistics of the Galactic thermal dust emission. Since the dust grains tend to align with their short axes parallel to the ambient magnetic field [12,13], the problem of CMB foreground removal is also entwined with the study of the Galactic magnetic field.

Planck mapped the full sky in nine frequency bands, of which seven are sensitive to polarization. In each polarization-sensitive frequency channel, Planck measures an intensity ($T$) map, a $Q$-polarization map and a $U$-polarization map. The $Q, U$ maps can be converted to a $E$-mode scalar map and a $B$-mode pseudo-scalar map. The statistics of the maps are often presented in the form of correlations between the $T, E, B$ components, i.e., the $TT, TE, TB, EE, BB, EB$ power spectra. The $T, E$ components are invariant under a parity transformation, whereas the $B$ component changes sign in the mirror world. Thus,

for models that preserve mirror symmetry, the ensemble averages of parity-odd $TB$ and $EB$ power spectra are expected to vanish. A detection of $TB$ or $EB$ correlation beyond statistical fluctuations, either in the Galactic foreground or in the CMB, would provide valuable information about parity-breaking physics beyond the standard picture [14–18].

The channel at 353 GHz, the highest polarization-sensitive frequency, is the most sensitive to the Galactic polarized dust emission. The Planck 353 GHz polarized sky map exhibits a few non-trivial properties, such as a significantly non-unity $EE/BB \sim 2$, a positive $TE$ correlation and a weakly positive $TB$ correlation [9,10,19]. All these features except for the parity-odd $TB$ signal can be qualitatively explained by state-of-the-art magneto-hydrodynamic (MHD) simulations [20–22] and by phenomenological modeling of the magnetized filamentary structure of the interstellar medium (ISM) [23,24]. In particular, the publicly available filament-based code DUSTFILAMENTS [24] is able to reproduce the $EE$, $BB$ and $TE$ power spectra, as well as some non-Gaussian features (Minkowski functionals) of the Planck 353 GHz sky map. The DUSTFILAMENTS code is also shown to be in good agreement with MHD simulations.

It has been demonstrated, with strong evidence, that the parity-odd $TB$ signal in Planck high-frequency maps is driven by misalignment between dust filaments and the sky-projected magnetic field [25,26]. However, a globally positive $TB$ correlation over a wide range of scales ($\ell \sim 40$–600), as seen in the Planck data, has only been reproduced in phenomenological simulations with artificial input of a significantly skewed distribution of the magnetic misalignment angles. For simple models with symmetric distribution of the magnetic misalignment angles, the $TB$ correlation arising from statistical fluctuations is typically much smaller than the observed one [24,25]. Thus, the observed $TB$ signal can be regarded as a hint about a global misalignment handedness preference. Quantification of the statistical significance of such a hint, the topic of the present work, is valuable for the exploration of the parity-breaking mechanism in the ISM beyond the current understanding of the Galactic physics [10,14,27].

The obvious strategy is to simulate dust emission maps with the null hypothesis of symmetric distribution of magnetic misalignment angles and to compare the $TB$ correlation of the Planck data with the correlations of the simulated maps . The problem of the direct comparison method is that even phenomenological filament-based simulations of Galactic dust emission are computationally expensive, and the dust $TB$ power spectrum may not obey a multi-variable Gaussian distribution. Straightforward detection of a $\sim 5\sigma$ anomaly in the $TB$ power spectrum would require at least$\sim$millions of simulated dust maps, which, taking the DUSTFILAMENTS code as an example, costs $\gtrsim 10^9$ CPU hours.

The present work aims to find a more computationally economical approach to estimating the statistical significance of the global misalignment handedness preference. The idea is to compress the information into a simple integrated quantity. Although the dust intensity and polarization maps are highly non-Gaussian, an integrated quantity contains the sums of many adjacent modes and therefore approximately obeys a Gaussian distribution whose parameters can be estimated with an affordably small set of simulations.

## 2. Data and Software

Following Section 3.3.1 of [9], we produce a series of nested "large region" (LR) masks: LR72, LR63, LR53, LR42, LR33 and LR24, where the postfix in LRmn represents the fraction (mn%) of the sky that is unmasked. To avoid possible bias from noise correlations, we compute the correlation between two different half-mission maps of Planck data release 3 [28]. We confirmed that all the results presented in this paper do not vary much if we use, for example, even–odd splitting of the data [28] or the SRoll maps with reduced large-scale systematic effects [29].

We use DUSTFILAMENTS, a code recently developed by the authors of [24], to simulate dust maps. DUSTFILAMENTS is a three-dimensional model composed of $\sim 10^8$ filaments that are imperfectly aligned with the magnetic field. Although the filament-only recipe is likely to be an oversimplification of the complex morphology of the ISM [30],

DUSTFILAMENTS is able to reproduce the main features of the Planck 353 GHz maps. In particular, it generates frequency decorrelation and non-Gaussian features on small scales that are both in agreement with the Planck data [24]. By default, we use the software settings suggested in Table 1 in [24], which assume a symmetric distribution of the magnetic misalignment angles, i.e., the null hypothesis that we wish to test.

All the maps are processed using the standard Healpix software [31]. Angular power spectra of masked maps are computed using NaMaster [32], whose $\ell$-bins are taken to be uniformly spaced with $\Delta\ell = 20$. Cross-power spectra between two different maps are always symmetrized. For instance, the $TB$ power spectrum $C_\ell^{TB}$ actually refers to $\frac{C_\ell^{TB}+C_\ell^{BT}}{2}$, according to the NaMaster convention.

## 3. Stacking the Dust Maps

Before giving a detailed statistical description, we wish to qualitatively demonstrate the anomalous $TB$ signal in the Planck data by stacking the polarization maps. The polarization stacking approach, which we sketch below, can be found in detail in [33,34].

With the flat-sky approximation, the spin-2 $(Q, U)$ component around a central pixel can be written as

$$Q \pm iU \approx \left(\partial_x \pm i\partial_y\right)^2 \nabla^{-2}(E \pm iB). \tag{1}$$

The $\nabla^{-2}$ operator is defined in Fourier space as a multiplier, $-\frac{1}{k^2}$. The local Cartesian coordinates $x, y$ are given by

$$x = 2\sin\frac{\theta}{2}\cos\phi, \ y = 2\sin\frac{\theta}{2}\sin\phi, \tag{2}$$

where $\theta$ is the angular distance from the central pixel and $\phi$ is the angle between the radial vector from the central pixel and the longitudinal vector (pointing to the south pole). We use $\omega \equiv 2\sin\frac{\theta}{2}$ instead of $\theta$ to approximate the radial distance, because the transformation given by Equation (2) preserves the pixel area from spherical sky to flat sky [1]. The $(Q_r, U_r)$ components are obtained by rotating the $(Q, U)$ components from the local $(\vec{e}_x, \vec{e}_y)$ Cartesian basis to the radial and tangential polar basis,

$$Q_r = Q\cos(2\phi) + U\sin(2\phi), \tag{3}$$
$$U_r = U\cos(2\phi) - Q\sin(2\phi). \tag{4}$$

The correlations between $Q_r, U_r$ and the intensity at the central pixel $T(0)$ are given by

$$\langle Q_r(\omega)T(0)\rangle = -\int \frac{\ell d\ell}{2\pi} C_\ell^{TE} J_2(\ell\omega), \tag{5}$$
$$\langle U_r(\omega)T(0)\rangle = -\int \frac{\ell d\ell}{2\pi} C_\ell^{TB} J_2(\ell\omega), \tag{6}$$

respectively. Here $C_\ell^{TE}$ and $C_\ell^{TB}$ are the $TE$ and $TB$ power spectra and $J_2$ is the Bessel function of the first kind of order 2. The extra minus signs on the right-hand side of Equations (5) and (6) arise from the Bessel integral $\frac{1}{2\pi}\int_0^{2\pi} e^{i(\ell\omega\cos\phi - 2\phi)}d\phi = -J_2(\ell\omega)$.

According to Equations (5) and (6), the patterns of stacked $Q_r, U_r$ around pixels with a skewed distribution of $T(0)$ explore the $TE$, $TB$ correlations, respectively. Because the $TB$ correlation has a much lower signal-to-noise ratio than the $TE$ correlation, the conventional approach used in the literature of stacking polarization maps around intensity peaks does not yield a recognizable pattern. The signal-to-noise ratio can be enhanced if we stack $U_r$ around many more hot pixels. To focus on the scales where the anomalous $TB$ signal is

found, we filter both the intensity map and the polarization maps with a $100 \lesssim \ell \leq 600$ band-pass filter, defined by

$$F(\ell) = \begin{cases} 0, & \text{if } \ell \leq 90 \text{ or } \ell > 600; \\ 1, & \text{if } 110 \leq \ell \leq 600; \\ \sin^2 \frac{(\ell-90)\pi}{40}, & \text{if } 90 < \ell < 110. \end{cases} \qquad (7)$$

In Figure 1, we show the stacked $Q_r$, $U_r$ patches around the hottest pixels that cover $5\% \times 4\pi$ steradian within the LR63 mask. The left panels are the Planck result, and the right panels are obtained using the DUSTFILAMENTS simulation in [24], which is publicly downloadable. The similar patterns of stacked $Q_r$ in the upper panels confirm that the simple filament-based approximation can reproduce the observed dust $TE$ correlation to a reasonably good accuracy. The rich ring pattern of stacked $U_r$ in the Planck maps, however, is seen in the DUSTFILAMENTS simulation. The much weaker $U_r$ signal in the DUSTFILAMENTS simulation (faint blue ring in the lower-right panel) arises from statistical fluctuations, or in cosmological terms, the cosmic variance of the particular simulation used here.

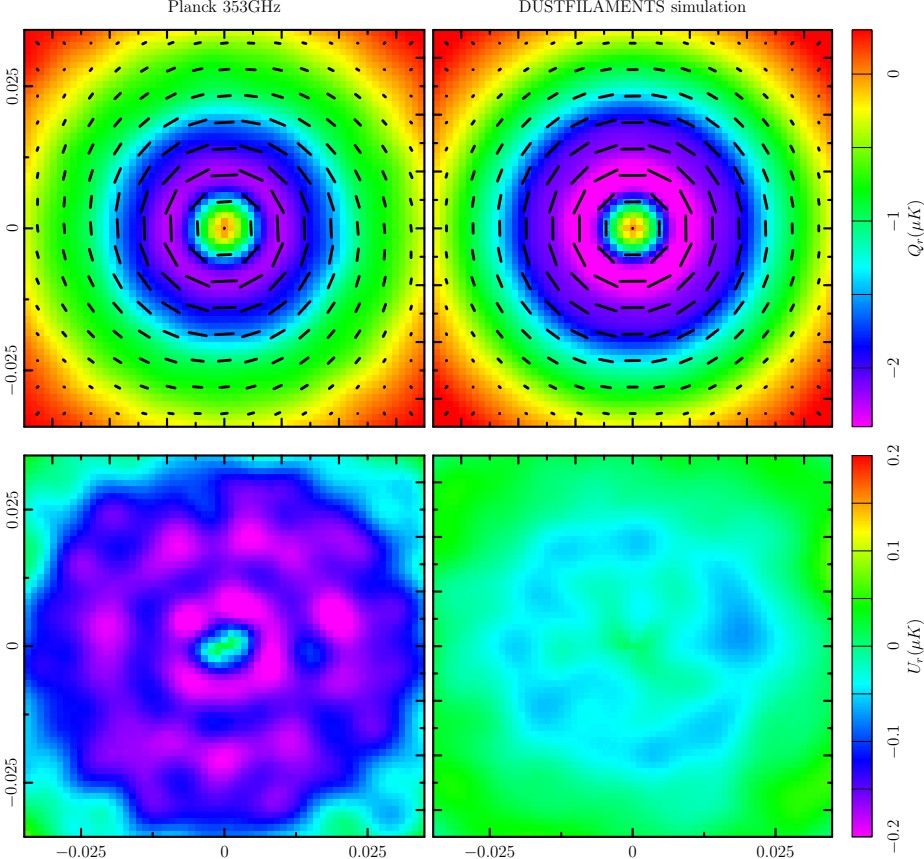

**Figure 1.** The stacked $[-2°, 2°] \times [-2°, 2°]$ ($[-0.035, 0.035] \times [-0.035, 0.035]$ in radians) patches of $Q_r$ (**upper** panels) and $U_r$ (**lower** panels) around the hottest pixels that cover $5\% \times 4\pi$ steradian in the LR63 mask. The colors represent the mean values, and the headless vectors in the upper panels show the polarization directions. The left column shows the Planck 353 GHz half-mission 2 ($100 \lesssim \ell \leq 600$ band-passed) polarization maps stacked around the hottest pixels of the Planck 353 GHz half-mission 1 ($100 \lesssim \ell \leq 600$ band-passed) intensity map. The right column shows the same for a DUSTFILAMENTS simulation.

The above example gives a good visual illustration that the typical $TB$ signal (stacked $U_r$) of DUSTFILAMENTS simulations is much smaller than the Planck observed one. The stacking approach, however, is not an optimal method for quantitative calculation of the

statistics. This is because a large set of simulations is needed to compute the non-diagonal covariance matrix and possibly the non-Gaussian features of the stacked $U_r(\omega)$ vector. Fourier-space modes are less coupled and hence are better for a statistical analysis, which we carry out below.

### 4. Statistical Analysis

Since we are mostly interested in a coherent nonzero $TB$ signal over a wide range of angular scales ($40 \lesssim \ell \lesssim 600$), a full likelihood analysis of the $TB$ power spectrum, which is computationally expensive, may be avoided by compressing the information to an integrated quantity

$$\xi_p \equiv \frac{\sum_{\ell=\ell_{\min}}^{\ell_{\max}} C_\ell^{TB} \ell^{-\alpha}}{\sum_{\ell=\ell_{\min}}^{\ell_{\max}} C_\ell^{TE} \ell^{-\alpha}}. \tag{8}$$

Unless otherwise stated, we fix the scale mask to be $[\ell_{\min}, \ell_{\max}] = [40, 600]$, the multipole range where the positive $TB$ signal was found in the Planck 353 GHz maps. The factor $\ell^{-\alpha}$, where $\alpha = 2.44$, roughly the spectral index of dust $TE$ and $TB$ power spectra found by the Planck mission [9,10], approximately sets equal weights for all multipoles. The weighted sum of $C_\ell^{TE}$ in the denominator makes $\xi_p$ insensitive to the overall normalizations of the simulated temperature and polarization maps. Note that the power spectra, and hence $\xi_p$, also depend on the sky mask, because the dust maps are not statistically isotropic.

According to the central limit theorem, the integrated quantity $\xi_p$ should approximately obey a Gaussian distribution. This greatly simplifies the problem, as we can then compute the probability density function $P(\xi_p)$ with a small set of simulations. Figure 2 shows the results from 36 DUSTFILAMENTS simulations with sky mask LR63 and scale mask $[\ell_{\min}, \ell_{\max}] = [40, 600]$. The Planck $\xi_p$ is about $16\sigma$ away from the simulated distribution, suggesting that simulations with symmetric distributions of magnetic misalignment angles fail to describe the reality.

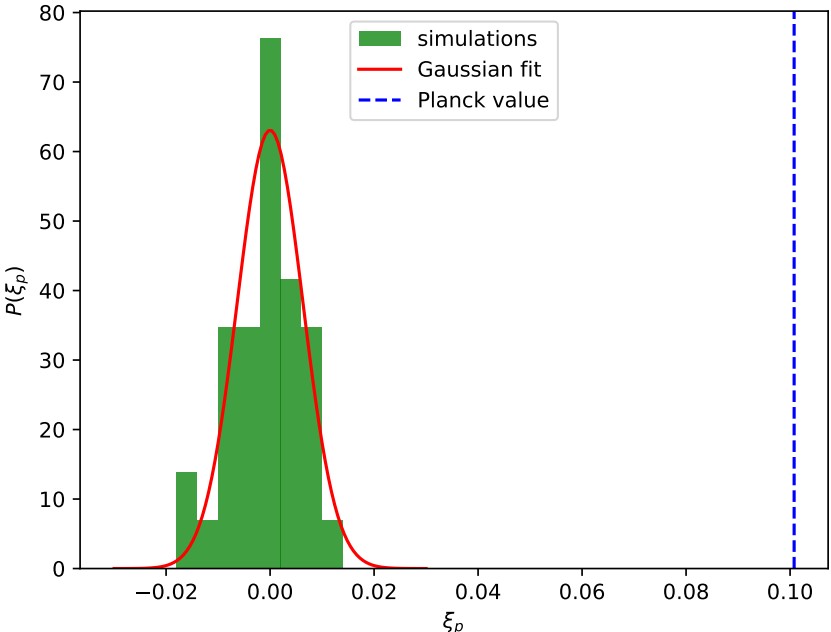

**Figure 2.** Gaussian fit to the histogram of LR63 masked $\xi_p$ from 36 DUSTFILAMENTS simulations. The measured Planck $\xi_p$ is in $15.9\sigma$ tension with the Gaussian fit.

To further test the look-elsewhere effect, we show in Table 1 the results for various choices of sky mask and scale mask. In all cases, the tension between the Planck data and simulations exceeds $10\sigma$, strongly rejecting the simulations as an acceptable description of the reality. We also list the excess kurtosis as a measure of the deviation from a Gaussian

tail distribution. For a comparison, in Figure 3, we show the distribution of the excess kurtosis of 36 samples drawn from a perfect Gaussian distribution. The kurtosis values in Table 1 are all in good agreement with the distribution shown in Figure 3, indicating that there is no evidence of non-Gaussianity in $P(\tilde{\xi}_p)$.

**Table 1.** The $\tilde{\xi}_p$ statistics from 36 DUSTFILAMENTS simulations. The last two columns show the Planck $\tilde{\xi}_p$ value and its tension with the simulations.

| Scale Mask $[\ell_{min}, \ell_{max}]$ | Sky Mask | Standard Deviation | Excess Kurtosis | Planck $\tilde{\xi}_p$ | Tension |
|---|---|---|---|---|---|
| [40, 600] | LR72 | 0.0054 | −0.98 | 0.068 | 12.6$\sigma$ |
|  | LR63 | 0.0063 | −0.13 | 0.101 | 15.9$\sigma$ |
|  | LR53 | 0.0072 | −0.33 | 0.091 | 12.6$\sigma$ |
|  | LR42 | 0.0080 | −0.94 | 0.138 | 17.3$\sigma$ |
|  | LR33 | 0.0084 | −0.43 | 0.199 | 23.8$\sigma$ |
|  | LR24 | 0.0096 | −0.48 | 0.183 | 19.2$\sigma$ |
| [80, 300] | LR63 | 0.0114 | −0.13 | 0.122 | 10.6$\sigma$ |

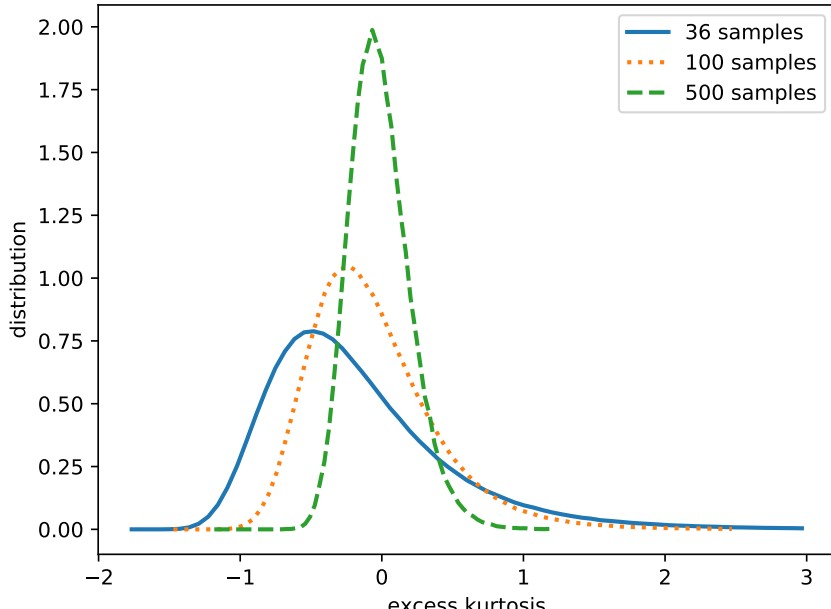

**Figure 3.** Distribution of excess kurtosis of a given number of random Gaussian samples.

Unlike the CMB statistics, which are known to be isotropic and very close to Gaussian, the statistics of Galactic dust emission are expected to be very non-Gaussian and spatially dependent. Although the central limit theorem guarantees that the integrated quantity $\tilde{\xi}_p$ approximately follows a Gaussian distribution, it is hard to quantify the level of proximity between $P(\tilde{\xi}_p)$ and a perfect Gaussian function. Ideally, one would also like to numerically test the convergence of $P(\tilde{\xi}_p)$ towards a Gaussian distribution by increasing the number of simulations. As shown in Figure 3, however, the distribution width of excess kurtosis does not shrink much if we increase the number of simulations from 36 to 100. Without investing much more computer power, we cannot tell whether the negative values of excess kurtosis in Table 1 are due to statistical fluctuations or are consequences of less-extended tail distributions of $\tilde{\xi}_p$. Nevertheless, the latter case (less-extended tails of $P(\tilde{\xi}_p)$) increases the tension between the data and the null hypothesis and therefore can only make our conclusion more robust.

## 5. Conclusions

The major conclusion of this work, that the observed dust $TB$ signal is inconsistent with a pure statistical fluctuation, may sound somewhat trivial in the sense that the cosmic variances (statistical fluctuations) of high-$\ell$ power spectra are all suppressed. However, as one can see from the lower-right panel of Figure 1, even the input model perfectly preserves parity symmetry, and a particular realization may still present recognizable $TB$ patterns at sub-degree scales. Thus, a quantitative estimation of the tail distribution of the $TB$ correlation is still necessary, especially when the observed dust $TB$ signal is small and the dust maps are highly non-Gaussian.

Since the null hypothesis of no preferred alignment handedness is ruled out, at least in a large neighborhood (∼a few hundred pc) around the solar system, there must exist a globally preferred handedness of the filament magnetic misalignment. Recent studies on the correlation between the Faraday depth and the polarization of the synchrotron emission suggest helicity in the large-scale Galactic magnetic field [27], which may lead to a dust $TB$ correlation [14]. However, this explanation has only been shown to be applicable to $TB$ correlation at large angular scales ($\ell \lesssim 50$) [14]. The physical origin of the parity violation at small scales (up to $\ell \sim 600$) remains unknown. Understanding the physical origin of this parity violation potentially has great value for many fields of astrophysics and cosmology. We also warn that the parity violation of the dust foreground must be carefully taken into account when studying the recently discovered tantalizing ∼2–3$\sigma$ hints of cosmic birefringence [15–18].

**Funding:** This work was supported by the National Key R&D Program of China (Grant No. 2020YFC2201600), the National Natural Science Foundation of China (NSFC) (Grant No. 12073088), the Guangdong Major Project of Basic and Applied Basic Research (Grant No. 2019B030302001) and the National SKA Program of China (Grant No. 2020SKA0110402).

**Data Availability Statement:** Planck maps are publicly available at https://pla.esac.esa.int/pla/, accessed on 28 June 2022.

**Conflicts of Interest:** The author declares no conflict of interest. The funders had no role in the design of the study, in the collection, analyses or interpretation of data, in the writing of the manuscript or in the decision to publish the results.

**Sample Availability:** Samples of the simulated dust maps are available from the authors upon request.

## Abbreviations

The following abbreviations are used in this manuscript:

| | |
|---|---|
| ΛCDM | Lambda cold dark matter |
| CMB | cosmic microwave background |
| MHD | magneto-hydrodynamic |
| ISM | interstellar medium |

## Note

1    For the $4° \times 4°$ stacking in the present work, the difference between $\theta$ and $2\sin\frac{\theta}{2}$ is actually unimportant.

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
