# Peer review of "The Galactic Interstellar Medium Has a Preferred Handedness of Magnetic Misalignment"

_universe, doi:10.3390/universe8080423_

Round 1

Reviewer 1 Report

The principal difficulty with this paper is that it is written for a very particular community, namely that around the Planck mission. I believe that ultimately one of that team should referee the paper. For a general reader, too much is introduced without explanation. For example the definitions of U, Q, E modes and B modes would benefit from some explanation. Particularly the resolution of U and Q along different orthogonal axes.

The nature of the physical effect is not made clear. I assume the field on the sky is detected in synchrotron polarization but how is this expected to be related to dust filaments? If these are distant and large enough to be affected by the galactic rotation, then perhaps a preferred helicity in the galactic magnetic field might be an explanation. The author should check the papers by Jennifer West et al. in MNRAS 507,4768, 2021 and 499,3673, 2020 for preferred global magnetic field behaviour.

In equation 1 the operator Delta^{-2} is not described.

figure 1 is not arranged in a left/right format so that its implications are unclear. 

Author Response

I would like to thank the referee's time and helpful suggestions. Below are my responses.

  1. "For a general reader, too much is introduced without explanation... " - This is an invited submission to the special issue "Cosmic Microwave Background". Thus, I am expecting the readers are familiar with Planck data and the conventions in CMB language. (I apologize if it is not the case, and I am ok to withdraw the submission and resubmit it to a  more appropriate journal.)
  2.  "The nature of the physical effect is not made clear. I assume the field on the sky is detected in synchrotron polarization but how is this expected to be related to dust filaments?..."  - The synchrotron emission are measured at much lower frequencies (e.g. WMAP 23 GHz map and Planck 30 GHz map). Here the Planck 350 GHz map is dominated by thermal dust emission. Dust emission is also polarized because dust grains are not spherical, and they have preferred alignment with the ambient magnetic field. Nevertheless, it is true that both the synchrotron map and dust map are related to the large-scale Galactic magnetic field. I added some comments (marked bold blue) in the conclusion section:   Recent studies on the correlation between Farady depth and polarization of the synchrotron emission suggest helicity in the large-scale Galactic magnetic field~\cite{West20}, which may lead to a dust $TB$ correlation~\cite{Bracco19}. However, this explanation only applies to $TB$ correlation on large angular scales ($\ell \lesssim 50$). The physical origin of the parity violation on small scales (up to $\ell\sim 600$) remains unknown.
  3.  "In equation 1 the operator Delta^{-2} is not described." -  I added the definition  (marked bold blue) below eq. 1:   The $\nabla^{-2}$ operator is defined in Fourier space as a multiplier $-\frac{1}{k^2}$.
  4. "figure 1 is not arranged in a left/right format so that its implications are unclear."  -  figure 1 is so arranged for comparison between Planck data (left column) and simulations (right column), as well as comparison between Q_r stacking (upper row) and U_r stacking (lower row). It is not the usual format but I think it is more logical.

Reviewer 2 Report

 This paper discusses the very important topics of the CMB polarisation, detected with Planck mission (E-mode), and especially positive correlation in TB (with B-mode) .

The Author used the routines of DustFilaments ( Hervías-Caimapo and Huffenberger, 2022) to simulate of the dust-filament polarised emission as on the Planck polarisation maps.

After comparison the TB correlation of the Planck data the null hypothesis of no preferred alignment handedness is ruled out, at least around a large neighborhood of the solar system, there

must exist a globally preferred handedness of the filament magnetic misalignment.

Indeed the physical origin of this parity violation potentially is very critical for interpretaion of Planck, BICEP and future ( CMB-S4?) data.

I think the probability density function is useful parameter for conclusion that modeling results strongly support that the Galactic filament misalignment has a preferred handedness, whose physical origin is yet to be identified.

Thus the results of the paper are new, useful, and perspective.

The "dust problem" became the main obstacle for  reliable detection of the B-mode in CMB studies, as we know from the BICEP2 results.   

Author Response

I would like to thank the referee for the very insightful comments. Below are my responses.

  1. "Indeed the physical origin of this parity violation potentially is very critical for interpretaiton of Planck, BICEP and future ( CMB-S4?) data."   - Yes it is indeed one of the CMB-S4 target to further explore the dust TB correlation seen in Planck.
  2. ''The 'dust problem' became the main obstacle for  reliable detection of the B-mode in CMB studies, as we know from the BICEP2 results." - Yes it is one of the motivation of this work, and this work is submitted to the special issue of CMB.
  3. Other changes: I added definition of $\nabla^{-2}$ below eq. 1, and some further discussion about the physical origin of the parity violation in the conclusion section. All marked with bold blue.

Round 2

Reviewer 1 Report

The paper has been clarified as to the scale of the problem.